# The Influence of Anthropometric Variables and Filtering Frequency on Center of Pressure Data

**DOI:** 10.3390/s23115105

**Published:** 2023-05-26

**Authors:** Jan Jens Koltermann, Philipp Floessel, Franziska Hammerschmidt, Alexander Carl Disch

**Affiliations:** 1Consulting Engineer for Metrology and Data Science, Bahnhofstraße 33, 03046 Cottbus, Germany; 2TU Dresden-University Hospital Carl Gustav Carus, University Center of Orthopedics, Trauma & Plastic Surgery, 01307 Dresden, Germany; 3TU Dresden-University Hospital Carl Gustav Carus, University Comprehensive Spine Center (UCSC), 01307 Dresden, Germany

**Keywords:** center of pressure, anthropometry, frequency analysis, power spectral density, CoP filtering

## Abstract

Good postural control is considered to be a key component of an active lifestyle, and numerous studies have investigated the Center of Pressure (CoP) as a way of identifying motor deficits. However, the optimal frequency range for assessing CoP variables and the effect of filtering on the relationships between anthropometric variables and CoP are unclear. The aim of this work is to show the relationship between anthropometric variables and different ways of filtering the CoP data. CoP was measured in 221 healthy volunteers using a KISTLER force plate in four different test conditions, both mono and bipedal. The results show no significant changes in the existing correlations of the anthropometric variable values over different filter frequencies between 10 Hz and 13 Hz. Therefore, the findings with regard to anthropometric influences on CoP, with a reasonable but less than ideal filtering of the data, can be applied to other study settings.

## 1. Introduction

Postural control is a fundamental human motor skill and an elementary component for maintaining an upright posture during static and dynamic processes [1,2]. The full quantification of postural performance is challenging due to the inherent complexity and close interaction between sensory perception and motor output. For this reason, a variety of different qualitative and quantitative methods have been established to assess postural stability, especially in the context of different pathologies [3,4,5]. The evaluation of CoP is particularly important for the diagnosis and rehabilitation of various conditions, including lower back pain, anterior cruciate ligament injuries, functional ankle instability, Parkinson’s disease, multiple sclerosis, and in stroke patients [6,7,8].

Quantitative recordings often involve the evaluation of parameters that describe the deflections of the CoP (Center of Pressure). In the upright, bipedal stance, the CoP is the central starting point of all ground reaction forces (in the transverse plane). Their course is analyzed with a force plate over a certain period of time [6,7]. The CoP trajectory enables the indirect quantification of postural competence based on the body sway in a steady stance. It can be assumed that the extent of body sway is directly related to postural performance. The quieter the stance, the better the individual balancing skills and neuromuscular control [8]. The increased fluctuation of the CoP indicates the greater instability of the center of mass, which has to be compensated for by corrective measures (horizontal ground reaction forces generated by muscular movements) [9].

Due to the complexity and high biological variability of the postural control loop, different methods and parameters have been established to describe the CoP motion. In addition to the methodological conditions, such as foot position, visual condition, and sampling duration [10], the type of data processing seems to have a considerable influence on the measurement result.

Furthermore, there are no standardized methods for recording the CoP curve, so the results differ depending on the filters, frequencies, and measuring times used. Thus, there is a multitude of scientific publications with different procedures and varying qualities of method descriptions.

The need for the standardization of static posturography was already pointed out in 1981 at the Symposium of Posturography in Kyoto [11], and was later confirmed in publications [12,13].

The discrepancies, however, result in the drawback that numerous studies are not comparable to each other, or only to a limited extent [14,15], making sufficient analysis difficult.

Regarding the questions of whether and how the CoP data of a force plate should be filtered, there is a wide range of opinions, which is also reflected in the variety of measurement methods and settings used [16]. A gold standard for CoP data collection does not exist. The only consensus so far is that a low-pass filter should always be applied to CoP values, and the Butterworth filter has emerged as one of the most commonly used filters [16].

The filtering of raw values is necessary because the measurement chain imposes a “noise” on the actual measurement signal. This noise expresses itself graphically in a multitude of high-frequency signals with low amplitude that are superimposed on the actual measurement signal.

The type of filters considered in this study are frequency filters characterized by individual behavior. The Butterworth filter corresponds to RC elements (analogous to the combination of a resistor and a capacitor in circuit technology), and is characterized by a passband attenuation of n × 20 dB per frequency decade, where n corresponds to the order of the filter. The attenuation at the cut-off frequency is approx. 3 dB, i.e., a signal with the cut-off frequency is attenuated to 1/√2 ≈ 0.707-fold of the original signal [16].

The aim of this work is to show the relationship between anthropometric variables and different methods of filtering the CoP data, particularly using force plates. The influence of filtering on the overall correlations will also be investigated.

The hypothesis is that even small deviations from the optimal filter frequency will have a significant impact on the results of the length of the CoP track. However, relationships between variables influencing the CoP track such as age, weight, and shoe size should not change the result significantly.

## 2. Materials and Methods

For this cross-sectional study, 221 healthy subjects from 18 to 90 years of age were included. The study population consisted of 116 males and 105 females.

Subjects were only included if they were clinically without pathological findings, answered “no” to all questions on the German Society for Sports Medicine and Prevention’s “Introductory Questionnaire for Health Risk Assessment in Athletes”, had no lower extremity injuries in the previous 12 months, and reported no acute or other back pain in the previous 3 months (PAR-Q questionnaire, DGSP). Back pain was assessed using the Korff Chronic Pain Grade Questionnaire [17].

The measurements were performed over 120 s. This is necessary to achieve a resolution in the spectrum of 0.02 Hz. All subjects who could not stand on one leg for a minimum of 120 s and who had other limitations of the musculoskeletal system (e.g., artificial joint replacement) as well as serious neurophysiological diseases (such as epilepsy), were excluded.

During the measurements, 35 subjects were excluded because they did not complete the task (standing ability < 120 s). With regard to the number of persons included, there were no significant differences between men and women, whereas there was a wide range in the age distribution.

The anthropometric data of the subjects is summarized in Table 1.

The absolute age distribution of the participants is shown in Figure 1. The relatively low proportion of subjects over 60 years of age can be explained by the inclusion/exclusion criteria, which required the absence of any pathological motor limitations or any artificial joint replacements.

The CoP was measured using a Kistler force measuring system (Type 9260AA) with a sampling rate of 1 kHz. The A/D converter used was a Redlab 1608 module (with a resolution of 16 bits). The raw data was recorded, and the COP or COP curve was determined using LabVIEW 2014 (National Instruments, Austin, TX, USA).

The data acquisition system was programmed to assess each person in a standardized order under the same conditions.

### 2.1. Method

The subjects were measured in a standardized order and under constant conditions using the following protocol:Monopedal left, eyes open (MONOL)Bipedal, eyes open (BIEO)Monopedal right, eyes open (MONOR)Bipedal, eyes closed (BIEC)

There was a 5-min rest period between each condition. The standardization of the sequence was necessary to be able to detect and evaluate any fatigue effects in the subsequent analysis. The recording time was limited to 120 s per measurement. If there were signs of increasing instability or loss of balance before this time had elapsed, the measurement was stopped, and the data recorded up to that point was saved but not included in the analysis. In the statistical evaluation, the results of the two-legged stance with eyes open were presented in the tables for clarity. The results of the other stance conditions in the protocol are comparable in their core statements. To ensure the comparability of the measured values, the first five seconds were excluded from the data analysis, as it can be assumed that stable measurement results only occur after the first five seconds.

To standardize the standing position, the standing surface of the force plate was optically marked to specify the positioning of the foot or feet, depending on the state of the measurement. In addition, an optical mark was placed at a distance of 3 m from the measuring device at a height of 1.7 m, on which the subjects had to fixate their eyes during measurement conditions 1 to 3.

### 2.2. Analysis

The software-supported analysis of the measurement recordings was carried out using LabVIEW 2020 from National Instruments (Austin, TX, USA). The position of the CoP or the course of the CoP over the measurement time was determined from the raw data of the force distributions. The length of the CoP track is calculated on the basis of the Pythagorean theorem and continuously applied between the X and Y coordinate increments.

As a central parameter of the CoP trajectory, the CoP length (the distance travelled by the CoP) was determined and used for further analysis. The results were presented as mean values of the total cohort. The significance level was set as α = 0.05.

The cut-off frequency for the Butterworth filter was calculated according to Koltermann et al. (2019) [18]. To determine the cut-off frequency, an averaged PSD spectrum was calculated from all measurements. The percentage of the power components that could be assigned to each existing frequency was then determined. Next, the percentage change of the power components was evaluated, and the point where the change was less than 0.1% was defined as the relevant frequency. Based on the available dataset, a relevant frequency of 6.5 Hz was determined. From the Nyquist theorem, it can be derived that the cut-off frequency (Fc) should be at least twice as large as the fundamental frequency. In the present case, this corresponds to Fc = 13 Hz, which was taken as the reference frequency and applied to all analyzed measurements. The calculated cut-off frequencies Fc = 13 Hz and Fc = 10 Hz were used for the observations. The results for 16 Hz are not shown here, as they are similar to 10 Hz. From the point of view of the processing of the CoP data, filtering with a too small cut-off frequency is critical, as more information is lost, and therefore the case Fc = 10 Hz is shown here. Subsequently, the raw data were then filtered with different cut-off frequencies between 3 and 50 Hz, and the CoP was calculated for each subject and each filter. Next, a t-test was used to test the extent to which the individual measurement series differed with different filtering. An equivalence test (TOST) was then used to check which filter series had the same mean value as the output filter frequency. An ANOVA was utilized to check whether the relationship of the anthropometric values to the measurement result changed with different filtering.

The filtering of the raw values was performed with the filters integrated in LabVIEW. The third order Butterworth filter with a variation of the cut-off frequencies was applied to the raw data. The statistical processing of the data was done with R 4.0.3 software. The group of subjects was divided into the following classes for statistical investigation: gender, weight, age, and shoe size.

## 3. Results

For a better classification of the overall group, the measurement results were first examined for basic statistical variables to exclude irregularities. For the following representations, the calculated CoP tracks were filtered with a Fc = 13 Hz. In Table 2, the mean values and standard deviation of the measurements are displayed separately by gender and age group.

Table 3 shows the results of the CoP track length in relation to gender and weight classes.

Table 4 summarizes the results of the CoP track length sorted by gender and shoe size.

Figure 2 displays the t-test results for the CoP track length obtained from different measurement conditions in relation to the variation of the cut-off frequency of the filter. The CoP track length for each value of Fc was compared to the CoP track length obtained with Fc = 13 Hz.

Below 5 Hz, all measurement conditions showed a *p*-value smaller than α = 0.05. The “bipedal stand” conditions provided a *p*-value smaller than α = 0.05 from 30 Hz. For the “monopedal stand” conditions, no α = 0.05 was found in the observation range up to 50 Hz.

To further validate and narrow down the results of the t-test, an equivalence test was also conducted. Table 5 shows the results of the equivalence test. For this purpose, it is necessary to determine a confidence interval prospectively. A best practice for choosing the interval is to use ±10% [19] of the mean. For this study, it was decided to set the confidence interval to a maximum of ±8% to ensure statistical relevance.

To check whether the filter design has an influence on the relationship of the external variables to the length result or changes these relationships between them, an ANOVA analysis was calculated on the results for filtering at 13 Hz (Table 6) and on the results at 10 Hz [7]. The results for “bipedal eyes closed”, “monopedal left”, and “monopedal right” conditions are shown in Appendix A.

Table 6 and Table 7 show that between the calculated *p*-values for 13Hz and 10 Hz, the differences are smaller than 1%. Therefore, it can be assumed at this point that there were no significant changes in the tested relationships of the values. The same applies to the power of the results.

## 4. Discussion

This study investigated the impact of filter frequency on the result of the CoP track length. To do this, the cut-off frequency of the filter was varied around the calculated optimal cut-off frequency. A statistical analysis using a *t*-test and a TOST equivalence test showed, under the specification of α = 0.05, that the mean values of the CoP track length for the test group differed significantly depending on condition. This is understandable, as 99% of all power components were found in the frequency spectrum between 0 and 6.5 Hz, leading to an optimal cut-off frequency of 13 Hz. Several studies have already shown that 95% of the CoP track’s frequency components are below 1 Hz [14,18,20]. A study by Koltermann et al. (2019) similarly showed that 93% of all frequency components were below 1 Hz, and 97% were below 5 Hz. To ensure that 95% of all frequency components lie below 1 Hz, one needs at least 120 s of acquisition time and the filter design described above. The TOST test showed that for all conditions, the results obtained with a cut-off frequency of 12 Hz and 14 Hz were statistically equivalent to those obtained with a 13 Hz filter [18].The ANOVA results of all conditions examined showed that the subjects in a “bipedal stance” with open eyes and in a “monopedal stance” on the right leg (at least 185 of the study participants are right-footed) had the least differences among the groups studied. This is due to a missing perturbation in the “bipedal stance“ and the subjects having the largest possible support surface. Similarly, all neural control circuits were optimally used in this stance. The same argumentation is valid for the one-legged stance on the right foot, as the right leg is usually the preferred one, leading to a better motor implementation of the task. The literature suggests that the difference between men and women may be due to the fact that women do gymnastic exercises more frequently, which leads to a slight motor and muscular advantage [20,21]. However, the query of exercise frequency as well as the type of exercise was not explicitly asked for from the subjects in this study.

For “bipedal eyes closed”, differentiation was possible in regard to age, gender and weight. This might a result of the lack of visual control, which leads to a higher demand for balance control from proprioception and the vestibular organ. In the case of a “monopedal left” stance, it can be assumed that the test subjects have poorer neuromuscular control on their non-dominant side, leading to a clearer discrimination of the results.

Another important finding of the study was that filter selection (between 10–13 Hz) had no significant influence on the basic effect relationships between different anthropometric data and the CoP track length.

An analysis of the measurement results for standing stability in a “bipedal stance” with open eyes showed that female subjects had a lower CoP track length with increasing weight, whereas in men, the CoP track length increased with increasing weight. This is in line with the results of Vahtrik et al. (2014), who explained a lower CoP track length in female patients by a higher body mass index, leading to a lower path velocity of the CoP track length [22]. However, Hue et al. (2007) reported reduced postural control in overweight men compared to normal weight men [23]. The increased CoP track length could be interpreted as an inability to synergistically modulate the three systems (visual, vestibular, and somatosensory) involved in maintaining balance [24,25]. Both interpretive approaches could be correct. Further work considering additional parameters, such as the frequency spectrum, could help in the interpretive analysis of the influence of body weight on the CoP track length.

Our results suggest (see Table 4) that changing the filter frequency from 12 Hz to 14 Hz, compared to 13 Hz, has no impact on the effect relationship between the shoe size and the CoP track length. However, foot size, and thus the supporting surface, can still be considered as a relevant biomechanical variable that influences stance stability. This correlation was previously described by Chiari et al. (2002), among others [26]. In their work, however, they assume that there is a disturbing factor in the measurement procedure, and propose a normalization method to level out this influence. The relevance of this factor should be considered depending on the research question, as subsequent studies have shown that the choice of footwear and the change in the supporting surface can also affect stance stability and the risk of falling [27,28].

On the other hand, the influence of age remains consistent across different filter frequencies. The results of the present study on “bipedal stance” with open eyes support the findings of previous studies, such as Dault et al. (2003), who investigated the adaptation of postural control in young and older people. This was in regard to the fulfilment of different postural requirements of steady standing, with and without visual feedback. The frequency analysis showed that only young subjects were able to reduce the amplitude of the body’s center of gravity sway with a simultaneous increase in frequency [29,30]. Furthermore, the measurement data of the current study, obtained in a “bipedal stance” with eyes closed, shows a clear increase of the CoP track length with increasing age. The data obtained in “bipedal stance” with open eyes shows the opposite trend: a decrease in the CoP track length with increasing age.

A possible explanation for this might be a reduction in sensorimotor abilities and perception with increasing age, especially with the removal of visual control during the task of standing steadily. This notion is supported by the study results of Yeh et al. (2015), who also found age-related changes in postural control, and a close link between this and sensorimotor feedback [31].

In addition, Yeh et al. (2015) and Dault et al. (2003) both found that the strategies for maintaining postural control change with increasing age. This is shown by changes in the structure of the CoP track length and its frequency content [29,31].

Regarding the achieved results, in analyzing postural control in the context of age-associated questions, CoP track length data should include frequency analyses. Future studies should pay attention to the fact that the analyzed frequencies [29], as well as the measurement duration, have a significant impact on the CoP track length in assessing physiological levels of postural control [32].

Some limitations of the present study should be acknowledged. The study only included healthy, symptom-free subjects, and thus we cannot draw conclusions about the relationship between anthropometric variables and filter frequency in special cohorts, such as athletes from different sports or patients with artificial joint replacements.

In this work, a third order Butterworth filter was used. From a technical point of view, the results are directly transferable to the Bessel filter. Considering the order of the filter, the results for higher orders would differ less for the third order filter than for the first and second order filters.

However, as no objective medical history was taken by a medical professional, it is possible that people with mild impairments or structural joint changes may have participated in the current study.

## 5. Conclusions

The results of the study can be useful in planning future research by providing information on measurement setups, equipment selection, and data processing, including filtering, in addition to the conducting of relevant comparisons.

The use of a third-order Butterworth filter with Fc = 13 Hz showed that the power density spectrum between 0 Hz and 5 Hz is well preserved and is available for further analysis. The recording time of 120 s should be chosen in order to achieve a sufficient weighting of the performances of the individual power densities. Due to the different physiological control mechanisms, the frequency range for data analysis should be adjusted according to the duration of the measurement. It was discovered that the correlation between the anthropometric data and the CoP track length was not influenced by the tested filters. This means that even though the data were not filtered with the cut-off frequency of Fc = 13 Hz, the results on the anthropometric influences on the CoP track length are still applicable to other study situations. However, it is still unclear whether these conditions influence the frequency spectrum. Therefore, future work should include the frequency spectrum in addition to the CoP trace in order to gain deeper insight into the postural pathomechanisms.

By only considering the total length of the CoP track compared to the influence of individual body parts or movement patterns, such as upper body movement or the first strategies in the hip the total length of the CoP track and the stance cannot be separated from each other. This individual separation requires a detailed study of the whole CoP track system based on different frequency components in order to evaluate the specific conduction capabilities of each subsystem and the effectiveness of specific training interventions.

## Figures and Tables

**Figure 1 sensors-23-05105-f001:**
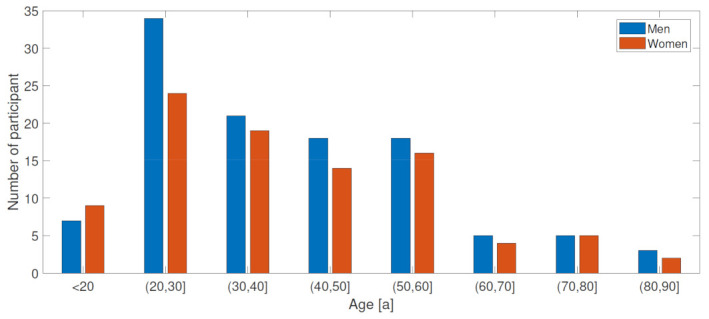
Age distribution of the subjects.

**Figure 2 sensors-23-05105-f002:**
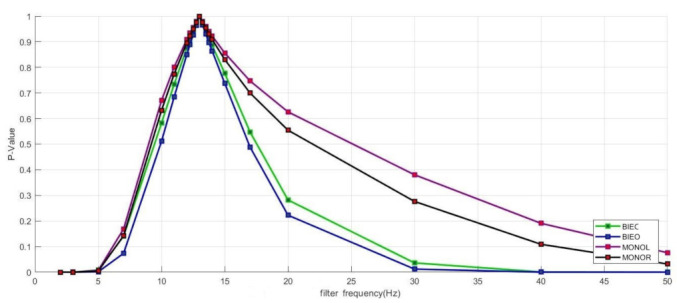
Overview of the t-test results for the comparison of the CoP track length mean values of the entire cohort at different filter frequencies to the output filter frequency of 13 Hz.

**Table 1 sensors-23-05105-t001:** Anthropometric data of the analyzed subjects, subdivided by gender.

	Total Group	Male (52%)	Female (48%)
Age [a]	39.9 ± 17.5	40.4 ± 16.6	40.5 ± 17.6
Mass [kg]	75.8 ± 15.6	83.9 ± 13.7	66.1 ± 11.9
Height [cm]	171.5 ± 25.8	180.9 ± 7.7	160.3 ± 34.2
BMI [kg/m^2^]	25.1 ± 4.2	25.6 ± 3.8	24.2 ± 4.5
Shoe size (EU)	42.8 ± 2.7	42.5 ± 5.5	38.9 ± 1.5
dominant side [%]	R 88/L 12	R 90/L 10	R 86/L 14
Subjects	186	98	88

**Table 2 sensors-23-05105-t002:** Overview of the CoP track length [cm] results separated by gender and age groups for filtering with 13 Hz.

Gender		Male				Female			
Classes_Age [a]		18	20–40	40–65	66	18	20–40	40–65	66
BI-pedal Eye Open	mean	146.24	153.33	131.35	135.94	151.08	167.91	138.01	118.98
	sd	27.78	70.57	42.31	47.18	30.34	76.506	62.18	27.38
BI-pedal Eye Closed	mean	166.57	152.9	173.27	342.15	142.43	147.67	143.16	175.05
	sd	50.76	59.32	61.42	242.60	48.42	49.90	44.51	94.33
Monopedal Right	mean	350.95	409.83	344.41	293.11	317.52	352.09	326.99	241.01
	sd	162.26	172.63	168.27	187.14	149.97	178.84	212.05	221.37
Monopedal Left	mean	368.24	405.70	332.99	284.27	328.65	327.25	264.57	213.28
	sd	154.37	190.06	165.47	178.96	138.64	139.13	173.11	175.07

**Table 3 sensors-23-05105-t003:** Overview of the CoP track length [cm] results separated by gender and weight classes for a filtering with 13 Hz.

Gender		Male			Female		
Class_Weight [kg]		60–80	80–100	>100	<59	60–80	80–100
BI-pedal Eye Open	mean	139.96	140.43	164.64	156.81	152.20	130.96
	sd	58.18	61.72	44.18	68.18	72.41	40.03
BI-pedal Eye Closed	mean	154.81	177.12	397.19	144.57	145.55	141.18
	sd	57.13	62.62	260.46	48.89	52.31	42.41
Monopedal Right	mean	366.67	392.66	277.60	360.05	316.81	283.38
	sd	175.09	175.06	122.86	162.10	220.64	118.60
Monopedal Left	mean	338.31	387.68	373.41	351.92	248.59	300.19
	sd	174.75	160.53	233.79	127.13	155.60	210.57

**Table 4 sensors-23-05105-t004:** Overview of the CoP track length [cm] results separated by gender and shoe size for a filtering with 13 Hz.

Gender		Male								
Shoe Size		39	40	41	42	43	44	45	46	47
BI-pedal Eye Open	mean	107.43	89.15	161.17	139.92	132.87	162.04	143.78	147.32	184.02
	sd	30.59	7.25	32.97	65.88	45.03	90.12	52.20	35.74	34.27
BI-pedal Eye Closed	mean	197.68	189.73	136.13	189.73	169.26	171.83	173.56	205.78	169.22
	sd	64.18	32.46	36.17	151.59	72.14	78.79	50.34	104.51	60.56
Monopedal Right	mean	284.51	345.27	367.90	379.43	364.68	395.10	353.63	350.14	404.92
	sd	105.04	156.04	210.49	174.00	167.24	224.92	149.57	217.78	22.12
Monopedal Left	mean	349.32	405.42	379.36	351.56	346.67	320.84	447.88	354.74	464.64
	sd	154.78	243.89	143.30	153.56	207.96	162.11	197.64	223.40	66.58
**Gender**		**Female**								
**Shoe Size**		**36**	**37**	**38**	**39**	**40**	**41**	**42**		
Bipedal eye open	mean	108.51	131.61	148.65	154.37	175.10	159.56	118.68		
	sd	49.54	44.70	66.52	71.66	61.17	84.78	31.11		
BI-pedal Eye Closed	mean	164.35	157.22	142.83	154.21	130.34	129.96	206.09		
	sd	37.45	35.65	40.96	58.92	48.19	47.99	127.18		
Monopedal Right	mean	443.79	317.42	384.87	325.80	258.34	274.71	188.88		
	sd	185.69	183.26	217.43	213.18	99.55	136.15	153.13		
Monopedal Left	mean	390.12	300.79	283.64	320.77	272.99	185.62	235.54		
	sd	131.04	140.00	157.77	183.03	124.75	142.16	140.74		

**Table 5 sensors-23-05105-t005:** Results for CoP track length of the TOST equivalence test with α = 0.05. Green indicates where the equivalence is significant in the comparison to those of Fc = 13 Hz.

Filter Frequency	BIEC	BIEO	MONOR	MONOL
10	0.08659	0.08105	0.8095	0.08099
11	0.05728	0.05083	0.05823	0.05912
12	0.03896	0.03045	0.04372	0.04521
14	0.03847	0.04013	0.03372	0.04447
15	0.05281	0.05695	0.04147	0.05329
17	0.09997	0.07504	0.05854	0.07113

**Table 6 sensors-23-05105-t006:** ANOVA for the measurements Bipedal eyes open, a Filter frequency of 13 Hz, and the anthropometric data.

Bipedal Eyes Open, Fc = 13 Hz	Statistic	*p*.Value	Etasq	Partial. etasq	Partial. omegasq	Power
Gender	0.503	0.48	0.003	0.005	−0.003	0.109
Shoe size	2.707	0.103	0.017	0.027	0.009	0.377
Classes Age	2.273	0.043	0.083	0.124	0.039	0.803
Class weight	0.421	0.833	0.013	0.021	−0.016	0.164
Gender:Shoe size	0.01	0.919	0	0	−0.005	0.051
Gender:Classes Age	0.85	0.535	0.031	0.05	−0.005	0.345
Shoe size:Classes age	0.414	0.868	0.015	0.025	−0.019	0.175
Gender:Class-Weight	0.118	0.949	0.002	0.004	−0.014	0.072
Shoe size:Class Weight	0.135	0.984	0.004	0.007	−0.024	0.082
Class_age:Class weight	0.604	0.929	0.096	0.141	−0.058	0.612
Gender: Shoe size: Class Age	0.838	0.544	0.031	0.05	−0.005	0.34
Gender: Shoe size: Class -Weight	0.279	0.841	0.005	0.009	−0.012	0.104
Gender: Class Age: Class- Weight	0.876	0.528	0.037	0.06	−0.005	0.39
Shoe_Size: Class-Age: Class-Weight	0.837	0.635	0.077	0.116	−0.013	0.594

**Table 7 sensors-23-05105-t007:** ANOVA analysis for the measurements “bipedal eyes open”, filter frequency of 10 Hz, and the anthropometric data.

Bipedal Eyes Open, Fc = 10 Hz	Statistic	*p*.Value	Etasq	Partial.etasq	Partial.omegasq	Power
Gender	0.416	0.521	0.003	0.004	−0.003	0.099
Shoe size	2.593	0.111	0.016	0.026	0.008	0.364
Classes-Age	2.146	0.051	0.079	0.118	0.035	0.776
Class-weight	0.418	0.835	0.013	0.021	−0.016	0.163
Gender:Shoe size	0.006	0.936	0	0	−0.005	0.051
Gender:Classes-Age	0.827	0.552	0.031	0.049	−0.006	0.335
Shoe size:Classes-age	0.406	0.873	0.015	0.025	−0.019	0.172
Gender:Class-Weight	0.098	0.961	0.002	0.003	−0.015	0.068
Shoe size:Class-Weight	0.113	0.989	0.003	0.006	−0.024	0.076
Class-age:Class-weight	0.602	0.93	0.097	0.14	−0.058	0.61
Gender: Shoe size: Class-Age	0.861	0.526	0.032	0.051	−0.004	0.35
Gender: Shoe size: Class-Weight	0.26	0.854	0.005	0.008	−0.012	0.1
Gender: Class-Age: Class-Weight	0.836	0.56	0.036	0.057	−0.006	0.372
Shoe-Size: Class-Age: Class-Weight	0.828	0.645	0.077	0.115	−0.014	0.588

## Data Availability

Not applicable.

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
