# Peer review of "The Influence of Anthropometric Variables and Filtering Frequency on Center of Pressure Data"

_sensors, 2023, doi:10.3390/s23115105_

Round 1
Reviewer 1 Report
This paper studies the influence of filtering frequency on sampled data, which is relatively simple and not innovative enough.​
Author Response
First, we would like to thank the reviewers for their careful and patient help in revising the manuscript.
Unfortunately, the comment is very short and we are not sure how to respond to it. The study we presented relates the influence of filters on the measurement data to the anthropometric data and examines possible interactions. To the best of our knowledge, this has not been done before and represents an further scientific value. Such investigations are particularly important in the discussion of the comparability of different methodologies in this field of research.
Reviewer 2 Report
This study investigated how filtering affects the results of a center of pressure analysis in healthy adults. The main finding of this study was that the 13 Hz was recommended as a cut-off frequency in a quiet postural task, and there were no significant changes in the relationship between the anthropometric values. While this study showed results with a huge cohort, I feel that this manuscript needs some more context for a better understanding.
My major concerns are as follows:
1. Table 1: Please change [a] to years. Also, it is the same as Fig. 1.
2. Line 123: The sentence “conditions 1 to 3” is a little unclear. Please change the pointed list to the numbered list from Line 105 to 108.
3. Method: Were all data acquired from one trial? If so, I think there were some subjects with a missing value. Please explain how to treat the missing values.
4. Line 150: Why do you use the 3rd order Butterworth filter in this study? I think that most previous studies applied the 4th Butterworth filter to the raw data from a force plate.
5. Line 158: There was not the variance of the measurements in Table 2. Please delete the sentence.
6. Some of the paragraphs should be written in the discussion such as Paragraph 2, 4 and 6
7. Line 301: Why do you think that the frequency range for data analysis should be adjusted depending on the measurement duration? I think this sentence is not followed by the current work or previous studies.
8. Please add the result of the dominant side in Table 1 (Anthropometric data). The data is needed for your discussion point (Line 241).
9. Please correct the name of the tables. Table 1 is duplicated.
10. All figures and tables don’t have captions. Please add it.
11. There are so many typos, grammar mistakes, and inadequate line breaks. I have listed a part of the mistakes in the minor points. Please re-check the whole manuscript.
Minor points:
1. Line 98: Please add [,] between “Instruments” and “Austin”.
2. Line 148: Please change “Anova” to “ANOVA”.
3. Line 157: Please add a space between “In” and “Table”.
Author Response
This study investigated how filtering affects the results of a center of pressure analysis in healthy adults. The main finding of this study was that the 13 Hz was recommended as a cut-off frequency in a quiet postural task, and there were no significant changes in the relationship between the anthropometric values. While this study showed results with a huge cohort, I feel that this manuscript needs some more context for a better understanding.
My major concerns are as follows:
- Table 1: Please change [a] to years. Also, it is the same as Fig. 1.
The correct unit description of years in the international system of units is [a] lat. annus
- Line 123: The sentence “conditions 1 to 3” is a little unclear. Please change the pointed list to the numbered list from Line 105 to 108.
we have revised the list
- Method: Were all data acquired from one trial? If so, I think there were some subjects with a missing value. Please explain how to treat the missing values.
If there were signs of increasing instability or loss of balance before the time had elapsed, the measurement was stopped, and the data recorded up to that point was saved but not included in the analysis
- Line 150: Why do you use the 3rdorder Butterworth filter in this study? I think that most previous studies applied the 4th Butterworth filter to the raw data from a force plate.
In the filter results, the actual difference between 3rd and 4th order is negligible. If you look at the frequency response in the diagram below, the difference in how steeply the curves are bent at the cut-off frequency is very small. We have also studied this in detail in Validation of Different Filters for Centre of Pressure Measurements by a Cross-Section Study. Technologies 2019, 7, 68. https://doi.org/10.3390/technologies7040068. Therefore, if the cut-off frequency is correctly determined, it does not matter whether the results are filtered with a 3rd or 4th order filter.
- Line 158: There was not the variance of the measurements in Table 2. Please delete the sentence.
we have revised the sentence - Some of the paragraphs should be written in the discussion such as Paragraph 2, 4 and
We have not implemented this suggestion. We think that the sections of the introduction are important for the introduction to the topic and fit better in this place in the text than in the discussion.
- Line 301: Why do you think that the frequency range for data analysis should be adjusted depending on the measurement duration? I think this sentence is not followed by the current work or previous studies.
we have rewritten the paragraph as follows;
Using a third-order Butterworth filter, the optimal analysis frequency spectrum was found to be between 1 Hz and 5 Hz, with a cut-off frequency of 13 Hz. Due to the different physiological control mechanisms, the cut-off frequency of the data analysis filter should be adjusted according to the range of the frequency spectrum to be investigated.
- Please add the result of the dominant side in Table 1 (Anthropometric data). The data is needed for your discussion point (Line 241).
we have added this information to the table
- Please correct the name of the tables. Table 1 is duplicated.
we have changed
- All figures and tables don’t have captions. Please add it.
All tables in the document have a heading, and figures have a caption.
- There are so many typos, grammar mistakes, and inadequate line breaks. I have listed a part of the mistakes in the minor points. Please re-check the whole manuscript.
we have revised the language of the document
Minor points:
- Line 98: Please add [,] between “Instruments” and “Austin”.
we have added - Line 148: Please change “Anova” to “ANOVA”.
we have changed
- Line 157: Please add a space between “In” and “Table”.
we have added
Reviewer 3 Report
Comments to the authors:
The authors attempt to study the relationship between the anthropometry variables and the filtering frequencies influencing the center of pressure data. The authors have extended their works in the following cited papers. From the title of the paper, it seems that the authors aim to use various filters, whereas they have used only a Butterworth filter where the frequency range has been varied. Such a title is misleading. However, they have presented all the anthropometric variables related to the CoP. The authors must suitably alter the title to showcase the work's actual content. One suggestion could be
“Influence of the anthropometric variables and filtering frequency on the Center of Pressure data”
[16] G. M. B. H. B. M. Koltermann JJ, "Validation of Various Filters and Sampling Parameters for a COP Analysis.," Technologies , vol. 6, p. 56, 2018.
[17] Koltermann, J.J.; Gerber, M.; Beck, H.; Beck, M, "Validation of Different Filters for Center of Pressure Measurements by a Cross-Section Study," Technologies, vol. 7, no. 68, 2019.
Author Response
The authors attempt to study the relationship between the anthropometry variables and the filtering frequencies influencing the center of pressure data. The authors have extended their works in the following cited papers. From the title of the paper, it seems that the authors aim to use various filters, whereas they have used only a Butterworth filter where the frequency range has been varied. Such a title is misleading. However, they have presented all the anthropometric variables related to the CoP. The authors must suitably alter the title to showcase the work's actual content. One suggestion could be
“Influence of the anthropometric variables and filtering frequency on the Center of Pressure data”
First of all, we would like to thank the reviewers for their careful and patient help in revising the manuscript.
-> we have adopted the title suggestion
Reviewer 4 Report
The study investigates “whether and how filtering affects the results of a posturograhic measurement, in particular the CoP analysis by means of force plates”, and “the influence of the filtering on overlapping correlations”. The authors concluded that there are constant conditions in the frequency range of 13Hz - 10Hz and no significant changes in the existing interrelationships of the values.
The results of this research are in my opinion of relevance to the field of sensors technology and would fit the scope of the journal. There are, however, minor concerns which should be addressed.
I would suggest to reformulate the title of the study.
Effect on the relationship between anthropometry variables and different ways of filtering the Center of Pressure data
I suggest removing the following sentences:
L10-11: Good postural control is considered as a key component in fall prevention. Therefore, numerous studies investigated the Center of Pressure (CoP) in standing position.
Instead, please add main findings. In addition, specify “constant conditions”.
L15-16: The study shows constant conditions in the frequency range of 13Hz - 10Hz and no significant changes in the existing interrelationships of the values.
The aim of the study should be also more precisely specified.
L13-14: The aim of this work is to provide frequency recommendations to increase the comparability of future CoP work.
Specify testing conditions (e.g., bipedal stance with eyes open).
L14-15: CoP was measured in 221 healthy volunteers using a KISTLER force plate.
The title of this manuscript says about “the relationship between anthropometry variables and different ways of filtering the Center of Pressure data“, whereas “anthropometry variables” are not mentioned in the aim of the study. Please, reformulate.
L68-70: The aim of this work is to show whether and how filtering affects the results of a posturograhic measurement, in particular the CoP analysis by means of force plates. In addition, the influence of the filtering on overlapping correlations is to be examined.
Specify inclusion criteria for subjects to be allocated to the study.
L77-79: In the context of this cross-sectional study, 221 healthy subjects, without limitations of the musculoskeletal system (without back pain), aged between 18 and 90 years were recruited.
Were patients included in the study?
L83: The anthropometric data of the patients is summarized in the following table.
Relevant limitations of this research should be discussed.
The conclusion should be considerably reformulated and shorthened. Main finding of the study should be only included. Citations of other works have to be removed from the conclusion.
L294-317: For future investigations, the current results could give assistance in planning measurement setups, choose necessary equipment, and give the chance to compare different study setups.
Future studies should pay attention to the fact that the analyzed frequencies [29] as well as the measurement duration have a significant impact on the CoP track in assessing physiological levels of postural control [32].
With a third-order Butterworth filter the optimal analysis frequency spectrum was demonstrated to be between 1Hz and 5Hz, with a cut-off frequency of 13Hz. Due to different physiological control mechanisms, the frequency range for data analysis should be adjusted depending on the measurement duration.
It is known that using the same frequency spectrum both, patients compared to healthy individuals and physiological training interventions in general, can influence the CoP track [33].
However, it is still unclear whether these conditions have an influence on the frequency spectrum. Therefore, in future work, in addition to the CoP track, the frequency spectrum should also be included in the consideration to obtain more in-depth information on postural pathomechanisms.
In addition, it is known from previous work, that the CoP track in standing position is the result of different subsystems of the individual to maintain standing stability [23]. However, the different influence of the subsystems on the total length of the CoP track in stance cannot be differentiated when only the CoP track is considered. To answer the specific conduction abilities of the individual subsystems and to prove the effectiveness of specific training interventions, future work should attempt to differentiate the total CoP track system-specifically on the basis of the different frequency components.
Author Response
The study investigates “whether and how filtering affects the results of a posturograhic measurement, in particular the CoP analysis by means of force plates”, and “the influence of the filtering on overlapping correlations”. The authors concluded that there are constant conditions in the frequency range of 13Hz - 10Hz and no significant changes in the existing interrelationships of the values.
The results of this research are in my opinion of relevance to the field of sensors technology and would fit the scope of the journal. There are, however, minor concerns which should be addressed.
First of all, we would like to thank the reviewers for their careful and patient help in revising the manuscript.
I would suggest to reformulate the title of the study.
Effect on the relationship between anthropometry variables and different ways of filtering the Center of Pressure data
we have reformulate the title of the paper
I suggest removing the following sentences:
L10-11: Good postural control is considered as a key component in fall prevention. Therefore, numerous studies investigated the Center of Pressure (CoP) in standing position.
We have rewritten the relevant sentences
Instead, please add main findings. In addition, specify “constant conditions”.
L15-16: The study shows constant conditions in the frequency range of 13Hz - 10Hz and no significant changes in the existing interrelationships of the values.
we have rewritten the part of the abstract as follows:
The study shows no significant changes in the existing correlations of the anthropometric variable values. over different filter frequencies between 10Hz and 13Hz. Therefore the findings on anthropometric influences on the CoP track, which were obtained with a less than ideal filtering of the data, can be transferred to other study settings.
The aim of the study should be also more precisely specified.
L13-14: The aim of this work is to provide frequency recommendations to increase the comparability of future CoP work.
we have revised the sentence on the objectives
Specify testing conditions (e.g., bipedal stance with eyes open).
L14-15: CoP was measured in 221 healthy volunteers using a KISTLER force plate.
We have included the suggestion in the text.
The title of this manuscript says about “the relationship between anthropometry variables and different ways of filtering the Center of Pressure data“, whereas “anthropometry variables” are not mentioned in the aim of the study. Please, reformulate.
L68-70: The aim of this work is to show whether and how filtering affects the results of a posturograhic measurement, in particular the CoP analysis by means of force plates. In addition, the influence of the filtering on overlapping correlations is to be examined.
The wording is imprecise and we have adapted it as suggested.
Specify inclusion criteria for subjects to be allocated to the study.
L77-79: In the context of this cross-sectional study, 221 healthy subjects, without limitations of the musculoskeletal system (without back pain), aged between 18 and 90 years were recruited.
We added inclusion criteria in the text
Were patients included in the study?
L83: The anthropometric data of the patients is summarized in the following table.
The wording is imprecise as patients been replaced by probands
Relevant limitations of this research should be discussed.
We have some relevant limitations in the discussion.
The conclusion should be considerably reformulated and shorthened. Main finding of the study should be only included. Citations of other works have to be removed from the conclusion.
we have adopted the advice and rewritten the conclution
Reviewer 5 Report
The manuscript entitled "Effect on the relationship between anthropometry variables 2 and different ways of filtering the Center of Pressure data" brings new and interesting information regarding the possibility to assess the control of posture, which is important in a variety of neurological disorders, affecting central and peripheral nervous system.
The following observation have to be made:
Introduction
Please give some diseases where s important to assess CoP for diagnosis and rehabilitation therapies results.
Material and method
Please write the exclusion criteria for your study group. Even you included healthy patients you need to mention how you eliminated the postural deficiency in your study group. Taking in consideration the age of your study group had even 90 y old patient (s) you need to explain this detail, because the control of posture in elderly can be disturbed by multiple neurological diseases (especially ischemic ones).
Conclusions
Please do not include references in Conclusions chapter. Is mandatory to exclude the text with references from this chapter. You should write only your conclusions based on your results. You should emphasize what is new in your study and how your results can improve the patient's condition.
Author Response
The manuscript entitled "Effect on the relationship between anthropometry variables 2 and different ways of filtering the Center of Pressure data" brings new and interesting information regarding the possibility to assess the control of posture, which is important in a variety of neurological disorders, affecting central and peripheral nervous system.
The following observation have to be made:
Introduction
Please give some diseases where s important to assess CoP for diagnosis and rehabilitation therapies results.
We have added in the introduction some diseases for which it is important to assess CoP data.
Material and method
Please write the exclusion criteria for your study group. Even you included healthy patients you need to mention how you eliminated the postural deficiency in your study group. Taking in consideration the age of your study group had even 90 y old patient (s) you need to explain this detail, because the control of posture in elderly can be disturbed by multiple neurological diseases (especially ischemic ones).
We have specified the exclusion criteria within the text. By excluding test subjects who were unable to stand on one leg for 120 seconds, we were able to avoid the problem of impaired control of postureindirectly, particularly due to ischemic diseases.
Conclusions
Please do not include references in Conclusions chapter. Is mandatory to exclude the text with references from this chapter. You should write only your conclusions based on your results. You should emphasize what is new in your study and how your results can improve the patient's condition.
we have adopted the advice and rewritten the conclution
Round 2
Reviewer 2 Report
Thank you for submitting your revised manuscript.
The authors have responded thoroughly to the comments and issues raised in the first review.
Author Response
thank you for the revirew. we are pleased to have fulfilled your comments to your satisfaction.